# Engineering Innate Immunity: Recent Advances and Future Directions for CAR-NK and CAR–Macrophage Therapies in Solid Tumors

**DOI:** 10.3390/cancers17142397

**Published:** 2025-07-19

**Authors:** Behzad Amoozgar, Ayrton Bangolo, Charlene Mansour, Daniel Elias, Abdifitah Mohamed, Danielle C. Thor, Syed Usman Ehsanullah, Hadrian Hoang-Vu Tran, Izage Kianifar Aguilar, Simcha Weissman

**Affiliations:** 1Department of Hematology and Oncology, John Theurer Cancer Center, Hackensack University Medical Center, Hackensack, NJ 07601, USA; 2Department of Internal Medicine, Rutgers New Jersey Medical School, Newark, NJ 07103, USA; 3Department of Internal Medicine, University of Washington School of Medicine, Seattle, WA 98195, USA; 4Department of Internal Medicine, Jefferson Health—New Jersey, Stratford, NJ 08084, USA; dthordo@gmail.com; 5RRH Lipson Cancer Institute, Rochester, NY 14621, USA; 6Department of Internal Medicine, Palisades Medical Center, North Bergen, NJ 07047, USA

**Keywords:** CAR-NK cells, CAR–macrophages, innate immunity, solid tumors, phagocytosis, cytokines, tumor microenvironment, adoptive cell therapy, iPSC, combinatorial immunotherapy, cellular therapy, hematology, oncology

## Abstract

This review highlights exciting new progress in using engineered innate immune cells—specifically CAR-NK (natural killer) cells and CAR–macrophages—to treat solid tumors. Unlike traditional CAR-T therapy, which has been successful in blood cancers but less so in solid tumors, CAR-NK and CAR–macrophage therapies can target tumors without relying on specific markers, infiltrate dense tumor environments, and activate broader immune responses. Recent innovations include improved cell design, the use of stem cells to create off-the-shelf therapies, and combination strategies with immune boosters or natural compounds like ginger and vitamin C. Early clinical trials have shown these therapies to be safe and promising, but further work is needed to enhance their durability, manufacturing efficiency, and tumor specificity.

## 1. Introduction

The clinical success of chimeric antigen receptor (CAR) T cell therapy in hematologic malignancies has galvanized the development of adoptive cellular immunotherapies. However, translating this success to solid tumors remains a formidable challenge due to the distinct biological features of solid cancers. Of these features, the tumor microenvironment (TME) imposes substantial physical and immunological barriers to effective CAR-T cell infiltration and function. Dense extracellular matrix components, abnormal vasculature, and stromal cell populations collectively restrict T cell trafficking and promote immune exclusion [1,2]. In parallel, the immunosuppressive milieu of solid tumors further undermines T cell activation and persistence through the presence of tumor-associated macrophages (TAMs), regulatory T cells (Tregs), myeloid-derived suppressor cells (MDSCs), and inhibitory cytokines such as TGF-β and IL-10 [3,4].

Another major limitation of CAR-T cell adaptation in solid tumors is antigen heterogeneity. Solid tumors often express target antigens in a spatially and temporally heterogeneous manner, thereby increasing the risk of immune escape following selective pressure from monovalent CAR targeting [5]. In contrast to hematologic malignancies, where lineage-specific markers such as CD19 and BCMA offer consistent targeting, many solid tumor-associated antigens (e.g., mesothelin, B7-H3, GPC3) are variably expressed and are not uniformly tumor-specific [6].

CAR-T cells also face intrinsic limitations when deployed against solid tumors. These include impaired tumor infiltration, limited persistence, and significant toxicity risks, including cytokine release syndrome (CRS) and Immune Effector Cell-Associated Neurotoxicity Syndrome (ICANS) [7,8]. Furthermore, the autologous nature of most current CAR-T manufacturing pipelines leads to variability in product quality, prolonged production timelines, and high costs. These barriers are particularly pronounced in the context of advanced solid tumors, where patients may have poor T cell fitness due to the cytotoxic effects of prior treatments [9].

Given these constraints, there is growing interest in harnessing innate immune cell platforms for CAR-based therapies. Natural killer (NK) cells and macrophages possess intrinsic tumor-homing capabilities, operate independently of MHC recognition, and exhibit superior safety profiles with reduced risk of graft-versus-host disease (GvHD) [10,11]. CAR-NK cells offer the ability to directly lyse tumor cells via granzyme B, perforin, and death receptor pathways, while also recruiting other immune effectors through cytokine secretion. In contrast, CAR–macrophages (CAR-MΦ) leverage phagocytosis, antigen presentation, and the ability to reprogram the TME through inflammatory cytokine production [12,13].

This review focuses exclusively on recent advances during the years 2023 to 2025 in the development of CAR-engineered NK cells and macrophages for the treatment of solid tumors. We highlight innovations in cell engineering, preclinical modeling, and early clinical translation, while also addressing persistent challenges such as in vivo persistence, antigen specificity, manufacturing, and microenvironmental resistance. Finally, we discuss emerging strategies to overcome these limitations, including dual-targeting CARs, immune checkpoint modulation, and the integration of pluripotent stem cell-derived platforms for scalable manufacturing.

## 2. CAR-NK Cells for Solid Tumors

### 2.1. Mechanisms of Action

CAR-engineered natural killer (NK) cells combine the synthetic targeting capacity of chimeric antigen receptors with the intrinsic cytotoxicity and immunoregulatory functions of NK cells. Unlike T cells, NK cells recognize and kill target cells in an MHC-unrestricted manner via a repertoire of activating and inhibitory germline-encoded receptors. Most notably, NKG2D, NKp30, and NKp46 detect stress-induced ligands or tumor-associated molecular patterns on malignant cells, enabling the targeting of tumors that evade adaptive immune recognition by downregulating MHC class I expression [14,15].

In addition to CAR-mediated cytotoxicity, NK cells exert direct antitumor activity through the release of cytotoxic granules containing perforin and granzyme B, which induce apoptosis in target cells. This is complemented by death-receptor-mediated mechanisms involving Fas ligand (FasL) and TNF-related apoptosis-inducing ligand (TRAIL) [16]. CAR-NK cells also secrete proinflammatory cytokines such as interferon-gamma (IFN-γ) and tumor necrosis factor-alpha (TNF-α), which not only contribute to direct tumor control but also modulate the tumor microenvironment (TME) to enhance antitumor immunity [17].

Importantly, NK cells can promote the recruitment and activation of other immune effector populations. Secretion of chemokines and cytokines by NK cells facilitates the recruitment of dendritic cells (DCs) into the TME, enhancing cross-presentation and priming of CD8^+^ T cells, thereby amplifying the systemic immune response against tumors [18,19]. This immunologic crosstalk between CAR-NK cells and components of the adaptive immune system may contribute to durable antitumor immunity in solid tumor settings.

### 2.2. Recent Advances: 2023–2025

Substantial progress has been made in recent years to enhance the efficacy and translational potential of CAR-NK cells for solid tumor therapy. A major innovation lies in the utilization of the overexpression of natural killer granule protein 7 (NKG7), a cytolytic effector molecule that plays a central role in granule exocytosis and immune synapse formation. In a 2024 study, Chen et al. demonstrated that NKG7 overexpression in B7–H3-targeted CAR-T cells enhanced IL-2 production, improved cytotoxic activity against solid tumor cell lines, and preserved CAR surface expression following repeated antigen exposure [20]. These findings will have direct implications for CAR-NK engineering in solid tumors and beyond.

Another area of active development involves optimizing intracellular signaling domains specific to NK cell biology. Unlike T cells, NK cells respond robustly to co-stimulation via adaptor proteins such as DAP10, DAP12, 2B4 (CD244), and DNAM-1 (CD226). CAR constructs that incorporate these domains have demonstrated superior degranulation, IFN-γ production, and persistence compared to canonical CD3ζ- or CD28-based designs. Li et al. showed that anti-mesothelin CAR-NK cells incorporating 2B4 and NKG2D domains displayed significantly enhanced cytotoxicity against ovarian cancer cells when compared to third-generation T-cell-based CARs [21]. Additional studies have also validated the utility of DNAM-1 and DAP12 in improving effector function and proliferative capacity of CAR-NK cells in solid tumor models [22,23].

Perhaps the most transformative platform development has been the generation of CAR-NK cells from induced pluripotent stem cells (iPSCs). These iPSC-derived NK cells (iNKs) offer a homogeneous, renewable, and scalable “off-the-shelf” source for CAR therapy. Crucially, they can be genetically engineered at early developmental stages to integrate synthetic enhancements such as chemokine receptors (e.g., CXCR2 or CCR7), checkpoint resistance modules, or cytokine support systems, thus ensuring improved trafficking, survival, and cytotoxic function in the immunosuppressive tumor microenvironment [24]. Li et al. demonstrated that iPSC-derived CAR-NK cells targeting CD19 maintained a stable phenotype and potent cytotoxicity both in vitro and in vivo, further supporting their feasibility for solid tumor adaptation [21].

These off-the-shelf products seek to address critical barriers posed by autologous manufacturing. Advanced solid tumor patients are often limited by lymphocyte exhaustion due to prior lines of cytotoxic therapy. When these patients are later considered for autologous processes, they face inconsistent product quality, delayed manufacturing, and reduced functional output. However, iPSC-derived or banked allogeneic CAR-NK cells aim to avoid these obstacles [25]. Together, these recent innovations in signaling design, cellular reprogramming, and synthetic augmentation are defining the next generation of CAR-NK platforms tailored for the unique challenges of solid tumor immunotherapy.

### 2.3. Preclinical Data

Several preclinical studies have highlighted the potential of CAR-NK cells in solid tumor models, particularly when engineered with lineage-specific enhancements to improve cytotoxicity, persistence, and resistance to immunosuppressive cues. A notable example is the incorporation of natural killer granule protein 7 (NKG7) into CAR constructs. In a recent study, Chen et al. demonstrated that ectopic NKG7 expression enhanced cytokine production (IL-2, TNF-α), sustained surface CAR expression, and promoted proliferation of B7–H3-targeted CAR-T cells in gastric and pancreatic cancer models, leading to significantly improved tumor control in the xenograft setting [20]. These findings offer a compelling rationale to apply NKG7 engineering to greater CAR-NK systems.

In addition to single-agent activity, combinatorial approaches are being actively explored. Oncolytic viruses have emerged as effective adjuvants, enhancing tumor immunogenicity, antigen presentation, and immune infiltration. In murine models of glioblastoma, the use of adeno-associated virus (AAV) vectors encoding checkpoint inhibitors in combination with HER2-targeted CAR-NK cells enhanced tumor clearance and promoted local inflammation within the tumor microenvironment [26]. These studies suggest that CAR-NK cells can be functionally augmented by virus-mediated immune reprogramming.

Checkpoint blockade strategies have also shown promise. A particularly innovative approach involves targeting the immunosuppressive PD-1/PD-L1 axis directly with CAR-NK cells. Robbins et al. engineered NK-92 cells to express a chimeric antigen receptor targeting PD-L1 and demonstrated potent cytotoxicity against human and murine head and neck squamous cell carcinoma (HNSCC) lines in vitro. In xenograft models, these PD-L1–CAR-NK cells significantly suppressed tumor growth, highlighting a strategy to convert an immune checkpoint molecule into a direct cytotoxic target [27]. This represents an important avenue for overcoming TME-induced dysfunction in PD-L1-rich tumors.

Collectively, these findings support the expanding therapeutic potential of CAR-NK cells in solid-tumor-directed cellular therapies. Preclinical success in gastric, pancreatic, glioblastoma, and HNSCC models provides a mechanistic basis for translation to early-phase trials through the context of rational combinations and immune evasion targeting.

### 2.4. Limitations

Despite this encouraging progress, CAR-NK cell therapies for solid tumors face several significant limitations that must be addressed to achieve consistent and durable clinical outcomes. A key biological limitation is the transient persistence of CAR-NK cells following infusion. Unlike T cells, NK cells typically do not undergo sustained in vivo expansion and often exhibit limited survival, even with exogenous cytokine support. This short lifespan can compromise their ability to exert long-term tumor control, particularly in solid tumors where immune escape and relapse are prevalent. To address this, several groups have explored strategies such as co-expression of IL-15 or membrane-bound cytokine complexes to promote autonomous NK cell persistence [10,28]. However, further experimentation is necessary to determine their in vitro efficacy.

Another major challenge is the inefficiency of viral transduction in primary NK cells. The known cytosolic pattern recognition receptors and antiviral restriction factors of primary NK cells generate intrinsic resistance to lentiviral and retroviral infection. Because of this, these cells are notoriously difficult to engineer using standard gene delivery systems. Recent studies have leveraged non-viral gene transfer approaches, such as the piggyBac transposon system, which enables stable co-expression of CARs and cytokine genes (e.g., IL-15) in human NK cells to improve their in vivo antitumor efficacy and persistence [29]. Additionally, the Mage transposon system has emerged as a novel non-viral gene delivery platform with high integration efficiency in mammalian cells, including immune effector populations such as NK cells [30].

Furthermore, CAR-NK therapies derived from peripheral blood or cord blood donors are subject to inter-donor heterogeneity in the NK cell receptor repertoire, expansion potential, and baseline functionality. These factors introduce variability into manufacturing and therapeutic potency. To mitigate this, there is increasing interest in the use of induced pluripotent stem cells (iPSCs) as a renewable and standardized source for CAR-NK generation. iPSC-derived NK (iNK) cells provide a genetically uniform platform suitable for off-the-shelf manufacturing, and recent studies have confirmed their scalability, cytotoxicity, and amenability to precise genome engineering [31,32]. These combined limitations underscore the need for ongoing innovation in CAR construct design, delivery systems, and manufacturing strategies to enable consistent, potent, and durable CAR-NK products for solid tumor cellular therapies.

### 2.5. Future Directions

To maximize the therapeutic potential of CAR-NK cells in solid tumors, several forward-looking strategies are under investigation to overcome limitations related to persistence, immune evasion, trafficking, and antigen heterogeneity. One promising avenue is the use of advanced gene editing technologies, such as CRISPR/Cas9, base editors, and transposon systems, to endow NK cells with enhanced persistence, immune evasion resistance, and metabolic fitness. For instance, genome editing has been used to knock out checkpoint receptors such as PD-1 or CISH to augment NK cell functionality in suppressive microenvironments [33]. Similarly, transposon-based delivery systems (e.g., piggyBac, Mage) have been employed to stably integrate CAR constructs alongside support genes like IL-15, thereby promoting a memory-like phenotype and reducing the need for systemic cytokine administration [29,30].

Another area of rapid development is the enhancement of NK cell trafficking to solid tumors. Engineering CAR-NK cells to express chemokine receptors such as CXCR2, CCR5, or CXCR1 has been shown to improve homing to inflamed tumor tissues, particularly in models of renal cell carcinoma and colorectal cancer [34]. These modifications aim to overcome the physical and chemotactic exclusion mechanisms characteristic of solid tumor stroma.

Furthermore, efforts are ongoing to broaden antigen recognition and reduce escape by designing dual-targeting CARs or universal CAR platforms. Tandem CARs, which incorporate two separate scFv domains against distinct tumor-associated antigens (e.g., B7-H3 and PD-L1), can simultaneously engage heterogeneous targets and reduce selective pressure [35]. Alternatively, universal CAR architectures such as those based on tag-specific adapters (e.g., FITC, biotin, or anti-tag antibodies) offer modular flexibility, allowing for real-time retargeting of effector cells against diverse antigens with a single cell product [36].

As the field progresses, these strategies are expected to converge into multi-functional, off-the-shelf CAR-NK products capable of circumventing the major barriers that have limited the success of cellular therapies in solid tumors. Future clinical trials will be critical in validating these approaches in terms of safety, efficacy, durability, and scalability.

## 3. CAR–Macrophages (CAR-MΦ) for Solid Tumors

### 3.1. Mechanisms of Action

Macrophages have emerged as a compelling platform for cellular immunotherapy in solid tumors due to their unique biology and intrinsic adaptability. Unlike lymphoid effectors, macrophages are abundant components of the tumor microenvironment (TME), often comprising up to 50% of tumor-infiltrating immune cells in epithelial malignancies [37]. They are naturally adept at penetrating dense stromal barriers and responding to chemotactic signals within the TME, where they can exert both tumor-promoting and tumor-suppressive roles depending on their polarization state [38].

Chimeric antigen receptor-engineered macrophages (CAR-MΦ) harness this intrinsic plasticity. Upon antigen engagement, CAR-MΦ can be reprogrammed toward an M1-like phenotype, associated with increased antigen presentation and expression of proinflammatory cytokines such as IL-12 and TNF-α and enhanced tumoricidal activity [13]. This contrasts with tumor-associated macrophages (TAMs), which are frequently skewed toward an M2-like immunosuppressive state in untreated tumors. Through synthetic reprogramming, CAR-MΦ offer the opportunity to reverse this immunosuppressive polarization and establish a proinflammatory, immune-permissive TME.

Mechanistically, CAR-MΦ exhibit a diverse antitumor repertoire. They mediate antibody-independent phagocytosis through CAR-driven cytoskeletal rearrangements, and they can also engage in antibody-dependent cellular phagocytosis (ADCP) when equipped with Fc-optimized CARs [12]. Additionally, trogocytosis enables CAR-MΦ to extract and degrade fragments of tumor membranes, resulting in antigen spreading and immune priming. CAR-MΦ also orchestrate broader immune responses by secreting chemokines that recruit T cells, NK cells, and dendritic cells, contributing to immune amplification within solid tumors [39,40].

Collectively, these properties make CAR-MΦ uniquely equipped to overcome several limitations of T and NK cell therapies in solid tumors, including stromal exclusion, poor antigen presentation, and resistance to apoptosis-inducing signals. Their ability to directly kill tumor cells, reshape the TME, and recruit other immune cells offers a multi-layered immunotherapeutic approach.

### 3.2. CAR-MΦ Design Features

The design of chimeric antigen receptors for macrophages requires substantial adaptations to align with myeloid-specific signaling. In contrast to lymphoid cells, macrophages rely on phagocytic receptor pathways rather than TCR- or CD28-based cascades. Accordingly, intracellular domains used in CAR-MΦ are derived from phagocytic and innate immune receptors.

Seminal studies have demonstrated that intracellular signaling motifs from Fc receptor gamma chain (FcRγ), DNAX-activating protein 12 (DAP12), and multiple EGF-like domains protein 10 (Megf10) are effective in driving macrophage activation. These domains engage SYK- and PI3K-dependent cascades to trigger actin remodeling and phagocytosis upon antigen engagement. Morrissey et al. showed that CARs incorporating FcRγ or Megf10 enabled antigen-specific engulfment and cytoskeletal polarization in murine and human macrophages [12]. Similarly, Klichinsky et al. and Paasch et al. confirmed that DAP12-based CAR-MΦ induced antigen-specific phagocytosis and inflammatory cytokine production [13,41].

To amplify proinflammatory signaling and overcome suppressive tumor environments, toll-like receptor (TLR) agonism has been used as an adjunct or built-in enhancer of CAR-MΦ activity. Activation of TLR pathways promotes macrophage polarization toward an M1 phenotype, characterized by secretion of IL-12, TNF-α, and type I interferons, thereby enhancing their tumoricidal and immune-recruiting capacities [42]. This has been observed in preclinical models of hepatocellular carcinoma, where TLR activation reversed TAM-mediated immune suppression [42].

Beyond single CAR constructs, new design strategies employ bicistronic or tandem systems to couple antigen-specific activation with payload delivery. Examples include CARs co-expressing IL-12, CD40L, or STING agonists, which reprogram the TME by enhancing antigen presentation and recruiting T and NK cell subsets [39,40]. These rational design choices are enabling CAR-MΦ to function not only as direct cytotoxic agents but also as orchestrators of local and systemic antitumor immunity.

### 3.3. Recent Advances: 2023–2025

Recent developments in the field of CAR-MΦ engineering have positioned them as a promising therapeutic platform for solid tumors, offering unique advantages over lymphoid-based approaches. Their innate ability to infiltrate the tumor microenvironment (TME), perform phagocytosis, and orchestrate immune responses makes them particularly well-suited for overcoming the immunologic and stromal barriers characteristic of solid malignancies.

A pivotal clinical milestone was achieved with the Phase I trial of CT-0508 (NCT04660929), an autologous anti-HER2 CAR-MΦ therapy. In this study, 14 patients with advanced HER2-overexpressing solid tumors, including breast and gastroesophageal cancers, were treated without lymphodepleting chemotherapy. CT-0508 manufacturing was successful in all patients, with high viability, CAR expression, and macrophage purity. The therapy demonstrated an excellent safety profile, with no dose-limiting toxicities. Importantly, 44% (four out of nine) of patients with HER2 3^+^ tumors achieved stable disease at 8 weeks post-treatment. Serial biopsies revealed CT-0508 trafficking into tumor tissue and evidence of TME remodeling, including expansion of CD8^+^ T cells and upregulation of genes involved in antigen presentation and interferon signaling [43].

In parallel, preclinical studies have reinforced the mechanistic potential of CAR-MΦ. Li et al. and Hadiloo et al. demonstrated that macrophages engineered with HER2 or mesothelin-specific CARs exhibited robust, antigen-dependent phagocytosis in murine models of ovarian, pancreatic, and breast cancer [39,40]. These CAR-MΦ secreted elevated levels of IL-12, TNF-α, and chemokines such as CXCL9 and CCL5, which contributed to local immune activation and recruitment of CD8^+^ T cells and natural killer (NK) cells into the tumor bed. Moreover, both studies showed that CAR-MΦ facilitated antigen spreading via tumor debris processing and direct antigen presentation, thus linking innate effector function with adaptive immune priming.

Zhang et al. further emphasized that macrophages, when appropriately reprogrammed, can reshape suppressive tumor ecosystems by polarizing toward a proinflammatory M1 phenotype and upregulating co-stimulatory molecules necessary for T cell engagement [41]. These features uniquely distinguish CAR-MΦ from CAR-T or CAR-NK therapies, expanding their therapeutic role beyond cytotoxicity to include immunologic coordination. Collectively, these findings highlight CAR-MΦ therapy as a rapidly maturing platform in solid tumor immunotherapy, distinguished by its capacity to remodel the TME and coordinate multifaceted immune responses.

### 3.4. Limitations

While chimeric antigen receptor macrophage (CAR-MΦ) therapies have shown promising activity in preclinical models and early clinical trials, several translational and mechanistic barriers currently limit their widespread clinical application. A central biological limitation is the low in vivo persistence of CAR-MΦ. Unlike CAR-T or CAR-NK cells, which can expand upon antigen engagement, macrophages are terminally differentiated and generally do not proliferate after infusion. Shah et al. demonstrated that although anti-PSCA CAR-MΦ exerted robust antitumor activity in pancreatic cancer models, their presence in vivo was transient and required repeat dosing to maintain the therapeutic effect [44]. This limited persistence constrains long-term immune surveillance and durable responses in solid tumors, especially in the absence of supporting cytokine signals in the tumor microenvironment.

Manufacturing complexity is another key challenge. CAR-MΦ are typically generated from autologous monocytes through ex vivo differentiation into macrophages, a process requiring strict adherence to cytokine-driven protocols and time-intensive workflows. Lu et al. and Chen et al. have highlighted that maintaining consistent cell identity, CAR expression, and inflammatory polarization across patient batches remains a major obstacle for Good Manufacturing Practice (GMP)-compliant production at scale [45,46]. Moreover, current platforms may lack scalability and affordability for broad clinical deployment, especially when compared to more established CAR-T or CAR-NK systems. Integration into these existing manufacturing systems may be a necessary engineering compromise to streamline the distribution of future CAR-MΦ products.

Another concern involves the potential for immune-related off-target inflammation. While macrophages are not inherently cytotoxic, CAR engagement—particularly in an M1-skewed phenotype—can trigger a significant release of inflammatory mediators such as IL-12, TNF-α, and type I interferons. As Chen et al. and Zhang et al. emphasize, these cytokines, although beneficial in reprogramming the tumor microenvironment (TME), may also lead to bystander tissue damage if macrophage activity is not properly controlled [41,46]. The risk is especially relevant in tumors with shared antigen expression in normal tissues or inflamed organs. Strategies to mitigate these risks include the incorporation of suicide genes, logic-gated CAR designs, and the use of inducible promoters or localized delivery mechanisms.

Overcoming limitations such as TME remodeling, antigen presentation, and immune recruitment will be essential for translating the unique advantages of CAR-MΦ into durable and safe therapies. Future research will need to optimize persistence, refine manufacturing protocols, and incorporate fail-safes to limit off-target effects.

### 3.5. Future Directions

Several innovative strategies are under development to overcome current translational challenges and fully harness the therapeutic potential of CAR-MΦ therapies in solid tumors. These approaches span advancements in manufacturing, tumor-selective activation, and combinatorial immunotherapy aimed at boosting the inflammatory potential of engineered macrophages.

A major technological leap is the exploration of induced pluripotent stem cell (iPSC)-derived macrophages as a renewable source for CAR-MΦ production. Unlike autologous monocyte-derived macrophages, iPSC-derived CAR-MΦ enable standardized, scalable, and genetically customizable manufacturing while meeting GMP standards. The ability to genetically engineer iPSCs at early stages allows for the introduction of transgenes prior to macrophage differentiation, such as cytokine support modules, chemokine receptors, or suicide switches [45]. In achieving this, manufacturers can improve the consistency and safety of the final therapeutic product.

Integrating suicide switches into iPSC-derived therapies is a critical safety strategy, allowing for the selective elimination of aberrant or residual pluripotent cells following administration of a small-molecule inducer [47,48]. One of the most established systems is the inducible caspase-9 (iCASP9) platform, which initiates apoptosis upon activation with a chemical dimerizer such as AP1903 [47]. In a foundational study, Shi et al. demonstrated that targeted insertion of iCASP9 into the AAVS1 safe harbor locus using a CAG promoter enabled robust and sustained expression throughout iPSC differentiation, including in mesenchymal stromal cells and chondrocytes [48]. Upon AP1903 exposure, these cells were rapidly and completely ablated in vitro and in vivo, thus validating the system’s utility for eliminating potentially tumorigenic cells [48]. Notably, their work revealed that the commonly used EF1α promoter was prone to silencing in iPSCs, whereas the CAG promoter supported consistent transgene activity [48]. Complementary reviews have emphasized the clinical relevance of the iCASP9 switch, citing its favorable kinetics, non-immunogenicity, and established track record in early-phase clinical trials as a controllable safety mechanism for cell therapies [47]. Together, these findings underscore the value of introducing suicide switches at the iPSC stage to ensure uniform transmission to differentiated lineages, such as macrophages, thereby enhancing both product safety and regulatory compliance.

Equally promising is the design of TME-responsive CARs that are selectively activated under pathological conditions such as hypoxia, acidosis, or oxidative stress. These “smart” CAR systems incorporate synthetic biology tools such as hypoxia-inducible promoters, pH-sensitive domains, or AND-gate logic circuits, which restrict macrophage activation to the hostile milieu of solid tumors. Such systems aim to reduce off-target effects and systemic inflammation by ensuring that proinflammatory macrophage functions are confined to the tumor site [46].

Hypoxia-responsive CAR systems offer a safety-enhancing strategy by restricting CAR expression to hypoxic regions of solid tumors, thereby reducing off-tumor activation and systemic inflammation [49]. Kosti et al. engineered a pan-ErbB-targeted CAR under hypoxia-sensing control, demonstrating selective CAR expression within hypoxic tumor xenografts and minimal activity in normoxic tissues [49]. Similarly, He et al. developed a hypoxia-inducible transcription amplification (HiTA) system that significantly curtailed CAR expression and cytotoxicity under normoxia while preserving potent antitumor functions in low-oxygen environments [50]. These studies substantiate that hypoxia-responsive promoter circuits can effectively confine macrophage activation to the hostile tumor microenvironment, thereby offering a promising safety mechanism for CAR-equipped cell therapies [49,50].

Another exciting area of progress is the combination of CAR-MΦ with intratumoral delivery of immunomodulatory agents such as STING agonists or TLR ligands. Chen et al. demonstrated that intratumoral injection of R848 (TLR7/8 agonist) and poly (I:C) (TLR3 agonist) synergistically enhanced antitumor immune responses in lung cancer models by reprogramming macrophage polarization toward an M1 phenotype and activating dendritic cells [51]. Similarly, Huo et al. showed that enforcing M1 polarization of CAR-MΦ further augmented their phagocytic activity, cytokine production, and recruitment of CD8^+^ T cells in solid tumor models [52]. These studies support the rationale for combining CAR-MΦ with local immune adjuvants to create a proinflammatory microenvironment that enhances both direct and indirect antitumor immunity.

Together, these emerging innovations point toward the next generation of programmable, tumor-specific, and immunologically potent CAR-MΦ platforms. The integration of iPSC technology, synthetic gene circuits, and combinatorial immunomodulation is expected to elevate the safety, precision, and efficacy of macrophage-based immunotherapy in solid tumors.

## 4. Dueling Perspectives on CAR-MΦ and CAR-NK Cells

### 4.1. Comparative Perspective: CAR-MΦ Versus CAR-NK

Innate immune-based chimeric antigen receptor therapies, such as CAR-NK cells and CAR-MΦ cells, are gaining traction as complementary approaches to conventional CAR-T therapies, especially for solid tumors. Their lineage-specific biology confers distinct mechanisms of cytotoxicity, immune recruitment, and tumor infiltration.

CAR-NK cells are cytotoxic innate lymphoid cells that eliminate tumor cells through the release of granzyme B, perforin, TRAIL, and Fas ligand [53]. Their tumor-homing potential can be enhanced via genetic engineering of chemokine receptors such as CXCR2 or CCR5 [54]. In contrast, CAR-MΦ derive from the myeloid lineage and utilize phagocytic mechanisms—including trogocytosis, antibody-dependent cellular phagocytosis (ADCP), and proinflammatory cytokine release—to mediate antitumor activity [45].

The intracellular signaling domains in CAR constructs are tailored to each lineage. CAR-NK platforms leverage domains such as CD3ζ, DAP10, DAP12, 2B4, and DNAM-1 to initiate NK-specific signaling cascades [22,55,56,57]. Meanwhile, CAR-MΦ employ phagocytic activation motifs like FcRγ, Megf10, and DAP12 to induce engulfment and inflammatory activation [45].

In terms of secretory output, CAR-NK cells predominantly release IFN-γ and TNF-α to exert cytotoxic effects and initiate secondary immune responses [53,55]. In contrast, CAR-MΦ exhibit a broader inflammatory signature, producing IL-12, TNF-α, CCL2, and CXCL9, which help reshape the TME and recruit T cells and dendritic cells [45,52].

Clinically, CAR-NK therapies have advanced into Phase I/II trials for both hematologic malignancies and emerging solid tumor targets (e.g., CD19, mesothelin), with a favorable safety profile and transient in vivo persistence [28]. CAR-MΦ, while still in early-phase development, are under evaluation in the landmark Phase I CT-0508 trial for HER2-overexpressing solid tumors [43].

Persistence remains a shared challenge. CAR-NK cells typically persist for weeks post-infusion with cytokine support but lack memory capabilities. CAR-MΦ, being terminally differentiated, exhibit limited in vivo durability and require additional engineering (e.g., IL-15 co-expression or iPSC sourcing) to extend their therapeutic window [45,52]. A generalized, comparative outline of CAR-MΦ and CAR-NK cells is available in Table 1 and Figure 1.

### 4.2. Future Opportunities and Combinatorial Strategies

Looking ahead, combinatorial and modular engineering strategies are being actively explored to optimize the antitumor efficacy, specificity, and persistence of CAR-NK and CAR-MΦ therapies. These strategies aim to address the complex immunobiology of solid tumors by leveraging cross-platform synergy, antigen targeting refinement, and localized immune activation.

One emerging avenue is the implementation of dual-effector therapies involving the sequential or concurrent administration of CAR-NK and CAR-MΦ cells. These platforms have complementary mechanisms: CAR-NK cells provide rapid cytotoxic activity via granule exocytosis, while CAR-MΦ reprogram the tumor microenvironment (TME) and promote sustained immune cell recruitment and antigen spreading. In preclinical glioma models, Look et al. showed that CAR-T, CAR-NK, and CAR-MΦ cells exhibit distinct but non-redundant activities, and their function is significantly enhanced by cytokine co-administration [58].

Engineered cross-talk mechanisms further enable synergy between platforms. For example, CAR-MΦ can be designed to secrete IL-15 or IL-12, thereby promoting the persistence, proliferation, and cytotoxic function of CAR-NK cells. Strassheimer et al. demonstrated that CAR-NK cell therapy, when combined with checkpoint inhibitors, induced a secondary innate-like immune response via NK-T cell recruitment, thus highlighting the potential for cooperative signaling in multi-effector systems [59].

To improve tumor selectivity and mitigate off-target effects, researchers are advancing bispecific CARs and logic-gated constructs. These include dual-target CARs that require recognition of two tumor-associated antigens (e.g., B7-H3 and GPC3, or EGFRvIII and mesothelin) or use AND-/NOT-gating logic to fine-tune activation thresholds. As highlighted by Peng et al., dual-antigen targeting is especially critical in solid tumors due to spatial antigen heterogeneity and immune escape [60].

In parallel, combinatorial delivery approaches are under investigation to locally reprogram the TME. One such approach involves the intratumoral administration of CAR-MΦ in combination with TLR ligands (e.g., R848, poly (I:C)) or STING agonists. These innate immune stimuli enhance M1 polarization of macrophages, stimulate dendritic cell activation, and recruit CD8^+^ T cells, effectively amplifying systemic antitumor immunity while minimizing systemic cytokine toxicity. Chen et al. recently showed that such combinations in murine lung tumor models synergistically promoted immune activation and tumor regression [51]. Together, these future directions reflect a shift toward next-generation immunotherapy designs that are modular, combinatorial, and tailored to the dynamic ecosystem of the solid tumor microenvironment.

One area of significant promise is nutritional immunotherapy. For instance, plant-derived bioactives such as ginger have demonstrated antitumor properties and immune-boosting effects. Ginger extract has been reviewed comprehensively as a chemopreventive and therapeutic agent in cancer [61]. In addition, high-dose vitamin C (ascorbic acid), when properly administered, has shown considerable potential as an adjunctive therapy in both autoimmune and onco-hematologic diseases [62]. Earlier foundational reviews by Klenner also highlighted the preventive and therapeutic roles of high-dose ascorbic acid [63]. Taken together, the incorporation of such adjunctive, low-toxicity strategies alongside CAR-based cellular therapies may offer additional immune support, reduce adverse events, and enhance treatment durability. As these innate cell-based therapies advance into later-stage trials, expanding the scope of combinatorial strategies to include nutritional and metabolic modulation may significantly improve patient outcomes in solid tumor immunotherapy.

## 5. Conclusions

Chimeric antigen receptor therapies built upon innate immune effectors, namely CAR-NK cells and CAR–macrophages (CAR-MΦ), represent an evolving frontier in solid tumor cellular therapy. These platforms leverage distinct biological advantages over traditional CAR-T cells, including MHC-unrestricted recognition, natural tumor infiltration, cytokine-driven immunomodulation, and reduced risk of cytokine release syndrome. Recent innovations in CAR construct design, such as incorporation of DAP12, 2B4, or FcRγ signaling domains, as well as synthetic control circuits and tumor microenvironment (TME)-responsive elements, have significantly improved the efficacy and safety of these potential therapies.

Despite their promise, both CAR-NK and CAR-MΦ therapies face key translational challenges, including limited in vivo persistence, manufacturing constraints, and the complexity of antigen heterogeneity in solid tumors. However, ongoing clinical progress, exemplified by CT-0508, and the integration of induced pluripotent stem cell (iPSC) platforms, cytokine-enhanced cross-support, and dual-effector combinations offer feasible paths to clinical scalability and durability (Table 2).

Looking ahead, the convergence of synthetic biology, cellular engineering, and combinatorial immunotherapy will likely define the next generation of solid tumor treatments. As these innate cell-based therapies advance into later-stage trials, their refinement and integration into multimodal regimens will be critical to expanding the impact of adoptive immunotherapy beyond hematologic malignancies and into the solid tumor domain.

## Figures and Tables

**Figure 1 cancers-17-02397-f001:**
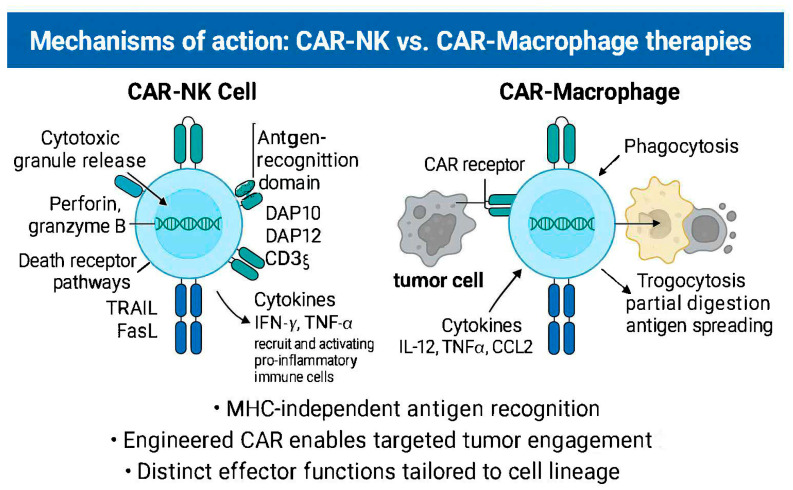
Mechanism of action: CAR-NK vs. CAR–macrophages.

**Table 1 cancers-17-02397-t001:** Comparative Features of CAR-NK vs. CAR-MΦ.

Feature	CAR-NK Cells	CAR–Macrophages (CAR-MΦ)
Lineage	Innate lymphoid lineage	Myeloid lineage
Mechanism of Cytotoxicity	Granule exocytosis (perforin, granzyme B), death receptor signaling (TRAIL, FasL), cytokine release	Phagocytosis, trogocytosis, antibody-dependent cellular phagocytosis (ADCP)
Tumor Infiltration	Limited natural infiltration; enhanced via chemokine receptor engineering (e.g., CXCR2, CCR5) [34,54]	Naturally efficient stromal penetration via chemotaxis and TME responsiveness [45]
CAR Signaling Domains	CD3ζ, DAP10, DAP12, 2B4, DNAM-1 [22,55,56,57]	FcRγ, DAP12, Megf10; Syk-/PI3K-dependent activation [45]
Secretory Profile	IFN-γ, TNF-α, GM-CSF; promotes recruitment and priming of adaptive immunity [53,55]	IL-12, TNF-α, CCL2, CXCL9; induces M1 polarization and immune recruitment [45,52]
Immune Recruitment Capacity	Indirect via cytokines and cross-talk with DCs and T cells [17,18,19]	Direct recruitment of CD8^+^ T cells, NK cells, DCs via chemokines [39,40]
Antigen Presentation	Limited	Active antigen presentation with potential for T cell priming and antigen spreading [41]
Clinical Stage	Phase I/II trials for hematologic malignancies and emerging solid tumor targets (e.g., mesothelin, HER2) [28]	Phase I (e.g., CT-0508 for HER2^+^ tumors); preclinical models for mesothelin, PSCA [43,44]
In Vivo Persistence	Transient (days to weeks), often supported by IL-15- or iPSC-derived platforms [25,28]	Very limited; no expansion, terminal differentiation; being enhanced with IL-15- and iPSC-based strategies [45,52]
Manufacturing Source	Peripheral blood, cord blood, iPSC-derived platforms under development [24,25]	Autologous monocytes; iPSC-derived macrophages emerging for off-the-shelf production [45]
Key Limitations	Short persistence, low transduction efficiency, inter-donor variability	Low durability, complex GMP manufacturing, cytokine-related toxicity risk

**Table 2 cancers-17-02397-t002:** Summary of CAR-NK and CAR–macrophage clinical trials.

Trial Name/ID	Cell Type	Target Antigen	Tumor Type	Study Phase	Key Findings
CT-0508 (NCT04660929)	CAR–Macrophage	HER2	HER2^+^ solid tumors	Phase I (clinical)	44% SD in HER2 3^+^ tumors; TME remodeling with CD8^+^ T cell expansion
PD-L1 CAR-NK Trial (preclinical)	CAR-NK	PD-L1	Head and neck squamous cell carcinoma	Preclinical	PD-L1 targeting enhanced tumor cytotoxicity and control in xenografts
iPSC-derived CAR-NK (preclinical)	CAR-NK	CD19 (model antigen)	Various solid tumors	Preclinical	Stable phenotype and cytotoxicity; potential for off-the-shelf use
Anti-PSCA CAR-MΦ (preclinical)	CAR–Macrophage	PSCA	Pancreatic cancer	Preclinical	Robust antitumor activity; required repeated dosing
HER2/Mesothelin CAR-MΦ (preclinical)	CAR–Macrophage	HER2/Mesothelin	Ovarian, breast, pancreatic	Preclinical	M1 polarization; enhanced phagocytosis and immune recruitment
Glioma CAR-NK/MΦ combo (preclinical)	CAR-NK and CAR–Macrophage	EGFRvIII, HER2	Glioblastoma	Preclinical	Distinct activity profiles; combination enhanced by cytokines

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
