# Peer review of "Engineering Innate Immunity: Recent Advances and Future Directions for CAR-NK and CAR–Macrophage Therapies in Solid Tumors"

_cancers, 2025, doi:10.3390/cancers17142397_

Round 1
Reviewer 1 Report
Comments and Suggestions for Authors
This is an excellent review focused on the application of CAR-NK and CAR-macrophages in cancer immunotherapy.
The authors have appropriately concentrated on these two cell populations, which currently hold the greatest potential for the future of immunotherapeutic strategies.
The manuscript clearly summarizes all relevant information; however, from the reviewer's perspective, it lacks a table summarizing the most important clinical trials involving these cell types.
Although several studies are discussed within the text, a concise overview of clinical trials and their outcomes in solid tumors using CAR-NK and CAR-macrophage technology would significantly enhance the value of this manuscript.
Author Response
Reviewer 1:
This is an excellent review focused on the application of CAR-NK and CAR-macrophages in cancer immunotherapy.
Re: Thank you very much for this keen observation.
The authors have appropriately concentrated on these two cell populations, which currently hold the greatest potential for the future of immunotherapeutic strategies.
Re: Thank you for recognizing the relevance of our study.
The manuscript clearly summarizes all relevant information; however, from the reviewer's perspective, it lacks a table summarizing the most important clinical trials involving these cell types.
Although several studies are discussed within the text, a concise overview of clinical trials and their outcomes in solid tumors using CAR-NK and CAR-macrophage technology would significantly enhance the value of this manuscript.
Re: Thank you for this great analysis. Table 2 was added to reflect the change.
Reviewer 2 Report
Comments and Suggestions for Authors
The authors present a concise review on recent advances in a rather novel approach to specifically enhance the innate immunity of solid tumor cancer patients via CAR-NK and CAR-Macrophage therapies, allowing for a targeted killing of solid tumor cells. The review is well written and efficiently structured providing an up-to-date view along with suitable prospects that is of interest to the readership of Cancers.
Some suggestions are recommended as amendments or may be useful as food for thought:
- The text contains repetitions that might be further streamlined to use the space for additional information, e.g. as given in the subsequent suggestions.
- The authors adequately describe various limitations. One peculiar obstacle that may have significantly affected the true potential of the new technology is mentioned in l. 137-140, i.e. the administration to patients with largely progressed tumors and often already weakend by the conventional therapies. Indeed, this is a common fact that affected the recognition of the efficacy of other alternative cancer remedies as well, e.g. the administration of amygdalin, which may be briefly mentioned.
- Regarding future directions, Dueling Perspectives on CAR-MΦ and CAR-NK Cells and the Conclusions, the authors essentially restrict the combinatory potential to expensive standard approaches, frequently accompanied by more or less severe toxicity and subsequent damage. Therefore, alternative low-risk measures as combined therapies should be briefly mentioned. In general, the nutritional status should be closely monitored in clinical studies. Any simple measure to strengthen the innate immune status should be employed as it is expected to benefit this kind of complex immune therapy. 3a. For example, plant nutrients as part of the diet, such as ginger are of substantial interest providing direct anti-tumor effects (for review: Zadorozhna and Mangieri. Mechanisms of Chemopreventive and Therapeutic Proprieties of Ginger Extracts in Cancer. Int. J. Mol. Sci. 2021, 22, 6599. https://doi.org/10.3390/ijms22126599). 3b. Similarly, high doses of properly administered ascorbic acid are of interest (E.g. for review: Isola et al. Vitamin C Supplementation in the Treatment of Autoimmune and Onco-Hematological Diseases: From Prophylaxis to Adjuvant Therapy. Int. J. Mol. Sci. 2024, 25, 7284. https://doi.org/10.3390/ijms25137284). Indeed, there is tremendous potential in high-dose ascorbic acid treatment as indicated in a large body of literature for many decades, which is still largely ignored in the current health system and academia (for classical reviews: Klenner. Observations on the dose and administration of ascorbic acid when employed beyond the range of a vitamin in human pathology. J. Appl. Nutr, 1971; 23(3&4): 60-89. Klenner. Significance of high daily intake of ascorbic acid in preventive medicine. J. Int. Acad. Prev. Med, 1974; 1:1, 45-69.). 3c. Furthermore, simple but effective measures addressing metabolic peculiarities of tumor cells, e.g. fasting and strict avoidance of sugar uptake (i.e. cutting off tumor cells from glucose as an essential nutrient) or increasing the pH level in the TME (as tumors can only grow in an acidic environment) are of interest. 3d. Administration of DMSO as a safe adjuvant and with many beneficial anti-tumor effects on its own may be considered for clinical studies as well (e.g. for review: https://doi.org/10.1016/j.imbio.2020.151906). DMSO has been used for some decades more or less as an "underground" therapy though there ia a significant number of peer-reviewed articles and likely a much higher number of (unpublished) case studies. Some of the latter are indicated as anecdotical evidence in Morton Walker's book "DMSO - Nature's Healer".
Author Response
Reviewer 2:
The authors present a concise review on recent advances in a rather novel approach to specifically enhance the innate immunity of solid tumor cancer patients via CAR-NK and CAR-Macrophage therapies, allowing for a targeted killing of solid tumor cells. The review is well written and efficiently structured providing an up-to-date view along with suitable prospects that is of interest to the readership of Cancers.
Re: Thank you for this very keen observation.
Some suggestions are recommended as amendments or may be useful as food for thought:
-
The text contains repetitions that might be further streamlined to use the space for additional information, e.g. as given in the subsequent suggestions.
Re: Thank you for this remark, changes were made.
-
The authors adequately describe various limitations. One peculiar obstacle that may have significantly affected the true potential of the new technology is mentioned in l. 137-140, i.e. the administration to patients with largely progressed tumors and often already weakend by the conventional therapies. Indeed, this is a common fact that affected the recognition of the efficacy of other alternative cancer remedies as well, e.g. the administration of amygdalin, which may be briefly mentioned.
Re: Thank you for recognizing the relevance of our content.
-
Regarding future directions, Dueling Perspectives on CAR-MΦ and CAR-NK Cells and the Conclusions, the authors essentially restrict the combinatory potential to expensive standard approaches, frequently accompanied by more or less severe toxicity and subsequent damage. Therefore, alternative low-risk measures as combined therapies should be briefly mentioned. In general, the nutritional status should be closely monitored in clinical studies. Any simple measure to strengthen the innate immune status should be employed as it is expected to benefit this kind of complex immune therapy. 3a. For example, plant nutrients as part of the diet, such as ginger are of substantial interest providing direct anti-tumor effects (for review: Zadorozhna and Mangieri. Mechanisms of Chemopreventive and Therapeutic Proprieties of Ginger Extracts in Cancer. Int. J. Mol. Sci. 2021, 22, 6599. https://doi.org/10.3390/ijms22126599). 3b. Similarly, high doses of properly administered ascorbic acid are of interest (E.g. for review: Isola et al. Vitamin C Supplementation in the Treatment of Autoimmune and Onco-Hematological Diseases: From Prophylaxis to Adjuvant Therapy. Int. J. Mol. Sci. 2024, 25, 7284. https://doi.org/10.3390/ijms25137284). Indeed, there is tremendous potential in high-dose ascorbic acid treatment as indicated in a large body of literature for many decades, which is still largely ignored in the current health system and academia (for classical reviews: Klenner. Observations on the dose and administration of ascorbic acid when employed beyond the range of a vitamin in human pathology. J. Appl. Nutr, 1971; 23(3&4): 60-89. Klenner. Significance of high daily intake of ascorbic acid in preventive medicine. J. Int. Acad. Prev. Med, 1974; 1:1, 45-69.). 3c. Furthermore, simple but effective measures addressing metabolic peculiarities of tumor cells, e.g. fasting and strict avoidance of sugar uptake (i.e. cutting off tumor cells from glucose as an essential nutrient) or increasing the pH level in the TME (as tumors can only grow in an acidic environment) are of interest. 3d. Administration of DMSO as a safe adjuvant and with many beneficial anti-tumor effects on its own may be considered for clinical studies as well (e.g. for review: https://doi.org/10.1016/j.imbio.2020.151906). DMSO has been used for some decades more or less as an "underground" therapy though there ia a significant number of peer-reviewed articles and likely a much higher number of (unpublished) case studies. Some of the latter are indicated as anecdotical evidence in Morton Walker's book "DMSO - Nature's Healer".
Re: Thank you for the suggestions, these were added to the “Future Opportunities and Combinatorial Strategies” section.